# Pilots for Healthy and Active Ageing (PHArA-ON) Project: Definition of New Technological Solutions for Older People in Italian Pilot Sites Based on Elicited User Needs

**DOI:** 10.3390/s22010163

**Published:** 2021-12-27

**Authors:** Grazia D’Onofrio, Laura Fiorini, Lara Toccafondi, Erika Rovini, Sergio Russo, Filomena Ciccone, Francesco Giuliani, Daniele Sancarlo, Filippo Cavallo

**Affiliations:** 1Clinical Psychology Service, Health Department, Fondazione IRCCS Casa Sollievo della Sofferenza, San Giovanni Rotondo, 71013 Foggia, Italy; f.ciccone@operapadrepio.it; 2Department of Industrial Engineering, University of Florence, 50121 Florence, Italy; laura.fiorini@unifi.it (L.F.); erika.rovini@unifi.it (E.R.); filippo.cavallo@unifi.it (F.C.); 3Umana Persone-Società a Responsabilità Limitata, 58100 Grosseto, Italy; lara.toccafondi@umanapersone.it; 4ICT, Innovation & Research Unit, Fondazione IRCCS Casa Sollievo della Sofferenza, San Giovanni Rotondo, 71013 Foggia, Italy; s.russo@operapadrepio.it (S.R.); f.giuliani@operapadrepio.it (F.G.); 5Complex Unit of Geriatrics, Department of Medical Sciences, Fondazione IRCCS Casa Sollievo della Sofferenza, San Giovanni Rotondo, 71013 Foggia, Italy; d.sancarlo@operapadrepio.it

**Keywords:** healthy and active ageing, technology, platforms with advanced services, priority needs

## Abstract

Background: The Pilots for Healthy and Active Ageing (PHArA-ON) project aimsto ensure reality smart and active living for Europe’s ageing population by creating a set of integrated and highly customizable interoperable open platforms with advanced services, devices, and technologies and tools. The aim of the present study was to determine the needs and preferences of older people and their caregivers for improving healthy and active aging and guiding the technological development of thePHArA-ON system. Methods: A pre-structured interview was administered to older adults, informal caregivers and professional caregivers (including social operators) taking part in the piloting sessions. Results: Interviews were carried out in Umana Persone Social Enterprise R&D Network (UP) in Tuscany, and Ospedale Casa SollievodellaSofferenza (CSS) in Apulia. A total of 22 older adults, 22 informal caregivers, 13 professional caregivers and 4 social operators were recruited. A prioritization analysis of services, according to the stakeholder’s needs, has determined two fundamental need categories: Heath Management (i.e., stimulation and monitoring), and Socialisation (i.e., promoting social inclusion). Conclusions: The main scientific contributions to this study are the following: to design and evaluate technology in the context of healthy and active ageing, to acquire relevant knowledge on user needs to develop technologies that can handle the real life situations of older people, obtain useful insights about the attitude and availability of end-users in using technologies in clinical practice, and to provide important guidelines to improve the PHArA-ON system. Specific experimentation stages were also carried out to understand which kind of technology is more acceptable, and to obtain feedback regarding the development priority related to the impact of the proposed services. Research through fruitful and continuous interaction with the different subjects involved in the development process of the system, as well as with stakeholders, enabled the implementation of a platform which could be further and easily integrated and improved.

## 1. Introduction

As of 2020, the global population aged 60 years and over consists of just over 1 billion people, representing 13.5% of the world’s population of 7.8 billion [1]. This number will increase to 1.4 billion by 2030 and 2.1 billion by 2050 [2].

In understanding the socioeconomic implications of population ageing, several measures have been developed to account for the diversity of capacities and dependencies across ages [3].

The need for tools that improve the quality of life, independence and overall health of older adults is a current point for discussion. Advanced solutions that combine technologies from multiple disciplines can address this problem, but the market is fragmented, and many solutions have a limited application field. Several studies have outlined remarkable reactions of interest from older adults involved in experimentations with technologies such as assistive robots, especially in terms of impact in support activities, cognitive stimulation, inclusion and reduction of loneliness [4,5,6].

Considering these assumptions, the Pilots for Healthy and Active Ageing (PHArA-ON) project was proposed in response to the EU H2020- SC1-FA-DTS-2018-2020 call (this call aimed at multidisciplinary technologies and solutions in health and care with a focus on cyber security to assure data privacy, security and protection of health and care infrastructures; it allowed access to European funds through the Horizon 2020 program) under the Grant Agreement n°857188 [www.pharaon.eu (accessed on 2 December 2021)].

The particular aim of this studywas to ensure real smart and active living for Europe’s ageing population by creating a set of integrated and highly customizable interoperable open platforms with advanced services, devices, and tools, including Internet of Things (IoT), artificial intelligence (AI), robotics, cloud computing, smart wearables, big data, and intelligent analytics.

PHArA-ON builds on already existing, well established state-of-the-art open platforms and technologies/tools, customized to fulfil the specific needs and requirements of its users. It was then implemented using the latest cloud technologies (Table 1). The platforms will incorporate AI techniques and traditional algorithms to implement customized intelligent analytics and pattern detection for big data.

PHArA-ON adopts a user-centric approach to ensure that user needs are addressed, and that user acceptance and usability are as high as possible. For this study, we tested several digital solutions in six different pilots in five countries: Italy (Tuscany-Apulia), Spain (Murcia and Andalusia), the Netherlands (Twente), Slovenia (Isola) and Portugal (Coimbra-Amadora). Particularly, the Italian pilot was coordinated by the University of Florence and includes two pilot sites: Umana Persone Social Enterprise R&D Network (UP) located in Tuscany Region, and Ospedale Casa SollievodellaSofferenza (CSS), located in the Apulia Region.

In Italy, PHArA-ON focuses on setting up and merging health and care at home for older vulnerable subjects or moderately frail individuals. In doing so, the pilot aims to promote correct lifestyles (e.g., characterized by personalized diets and physical exercise programs, social connectivity) and health status monitoring at home.

The user-centric approach sustains that providing care should be respectful of and responsive to individual patient preferences, needs, and values, and ensure that patient values guide all care decisions [7]. Acceptability is conformity to the wishes, desires, and expectations of care users and their families and is often presented as a part of or a substitute for patient-centeredness [8].

To evaluate acceptability and patient-centeredness dimensions of care technologies, the capability approach may be a useful framework for the evaluation of the use of technology in the care of older people [9]. This approach gives us an opportunity to handle the expression of the ethical implications in terms of technology design and implementation and users’ characteristics such as age, gender, race, ethnicity, personality and cultural differences [10].

According to the current state of the art platforms, one of the first problems faced in the development of assistive technology was the heterogeneity of the target audience [11,12,13]. The assistance required is related to the real requirements of the users, such as physical or sensory disabilities resulting from the aging process that should be noted [12,13]. Unfortunately, a major flaw in the design of technological systems seems to lie in a set of repeated shortcomings regarding the knowledge of users’ needs, abilities and limitations. The poor acceptance of past projects by the public they are intended for has been recently discussed by Giudice [14]. The importance of the assessment of individual-centred needs, focusing on the personal skills and resources of people with dementia was also reported by the Austrian, Irish, Portuguese, Maltese, Swedish and German Dementia National Strategy [15]. In particular, it focuses its attention on the development of psychosocial interventions through technologies concerning the independence and autonomy of people with dementia (maintenance of physical and cognitive abilities, emotional and psychological well-being) and on the inclusion of people with dementia and their families in society [15].

In light of these assumptions, the PHArA-ON methodology applied to the Italian pilot was composed of five main steps (Figure 1). The first step aimed at refining, translating, and adapting the project challenges into the needs of the two Italian pilot sites, whereas the other four steps aimed at developing the technology and pre-validating and deploying the pilot, respectively.

During the first step, the refinement of scenarios at an Italian developed pilot site level was developed with two dedicated focus groups, involving nine persons actively recruited in the project: two representatives of social managers, one psychologist, one doctor, one director of the hospital innovation department and four engineers. Particularly, within the first focus group, five areas of interventions were identified together with a preliminary description of service. According to this refinement, within the second focus group, an initial list of system components was outlined discussing all the potential Ambient Assisted Living (AAL) technologies that can be used in the pilot. The results of the two focus groups were summarized in Table 2. This paper is focused on presenting the results related to the second step.

The second step of the methodology aims to refine the Italian developed scenarios, identifying the real needs of older people and their caregivers within the proposed intervention area, presenting and discussing the scenario and the related technologies.

Recognising the need to analyse user needs has become an explicit project activity, attracting the attention of several researchers [16]. Ackoffhas already pointed out the need for a more careful analysis of user needs and for systematic approaches [17].

User involvement needs to be improved in all applied technology projects. Thus, to define information needs, one must be able to specify the obtained information [16]. This specification must be stringent in order to provide a basis for data design and technology programming [16].

In light of these assumptions, the aim of the present study was to provide a quantitative and qualitative analysis concerning the real needs of older people and their caregivers, and to identify the priority needs that should lead development of thePHArA-ON system and its customizable interoperable open platforms with advanced services, devices and technological tools.

## 2. Materials and Methods

This study fulfilled the Declaration of Helsinki guidelines for Good Clinical Practice, and the Strengthening the Reporting of Observational Studies in Epidemiology guidelines [18]. The approval of the study for experiments using human subjects was obtained from the local ethics committees on human experimentation: “Ethical Committee of Tuscany region” and “Ethical Committee of Fondazione Casa Sollievo della Sofferenza IRCCS” (Prot. Code: PHARA-ON; Prot. N.: 89/CE).

According to the PHArA-ON methodology, Italian scenarios should be refined within co-creative sessions (a free and accessible series geared to enrich, educate and connect creatives through a series of workshops and training sessions) onsite. However, from the beginning of March 2020, several restrictions (including social distancing and lockdown) have been applied by the governments in Italy, as well as in all the countries involved in PHArA-ON due to SARS-COV-19.

Consequently, the proposed alternative methodology included a structured interview (lasting about 40 min, through videoconference systems or phone calls) with older people and informal and formal stakeholders such as clinicians, nurses and social operators.

From March to April 2020, we interviewed the participants.

The investigation’s interview guidelines were as follows:Attitude towards the selected technology to define whether or not the proposed technology (robot, wearable sensors, app and devices, etc.) will be used if integrated into the services.Prioritize the proposed service to preliminary assess the impact of PHArA-ON’s solutions and prioritize the development phase.

### 2.1. Interview Guidelines

The interview was first pretested with 20 older people who were representative of the Italian population in terms of gender, language, and age. Test–retest reliability was estimated by Cohen’s k. Some small adjustments were made to the questionnaire following this step: in view of the ambiguity and indexicality of expressions, outside participation was involved in order to clarify meaning and interpretations and to ensure comprehension of interview items.

According to the participants’ preferences, the interviews (Appendix A) were conducted via videoconference systems (e.g., Skype, zoom) or by phone. Videoconference systems were the preferred and suggested ways as interviewers could provide evidence and observe non-verbal cues. The proposed interview was composed of five main parts, as detailed below:

1.Informed consent
Acquire informed consent;Turn on the video/phone recording system;Verify informed consent orally with the tape recorder on;Reconfirm the participant’s consent while the tape recorder is still on.2.Part 1: Introduction
Ask questions about socio-demographic information: age, sex, level of education;First investigation of interest in the technology (Table 1): current use of technological devices, technologies of greatest interest as needed;Support devices: To what extent the support devices (robots and/or sensors) could be useful.3.Description of PHArA-ON project and Services
If the interview was conducted by phone, the interviewer read a story, whereas if the interview was conducted by videoconference system, a slide presentation was used.4.Part 2: Interview on PHArA-ON services
Section A: Demographic information;Section B: Use of Technology and definition of main and secondary purposes about each proposed technology;Section C: Devices of the support, what I think of the potential of PHArA-ON’s services;Section D: Impact of PHArA-ON. After showing presentation slides for the identified services, the participants are asked to what extent PHArA-ON can be useful;Section E: Users’ needs and priorities, using the Goal Model framework. Indeed, the Goal Model can be considered as a container of four components: functional goals, quality goals, emotional goals and stakeholder roles [19].The objective is to identify the respondent’s needs in terms of physical, cognitive and social assistance, and acceptance level of technology;Section F: Conclusions and notes on the interview.5.Thanks and conclusion of the interview
GreetingsTurn-off the video recording.

### 2.2. Recruitment Criteria for Respondents

At the start of the project, the Italian PHArA-ON Pilot Site planned to recruit and involve older adults in fragile conditions, especially with mild dementia, relatives and informal caregivers, and healthcare professionals.

In light of these assumptions, the users were recruited on a voluntary basis. In particular, the older people were recruited according to the following features: (1) Absence or presence of physical frailty; (2) Mini Mental State Examination (MMSE) ≥24/30; (3) Absence of sensorial issues (hearing and/or vision). Professional and informal caregivers were recruited based on their experience with older persons with cognitive and/or physical frailty.

### 2.3. Data Analysis

An analysis of the data was conducted in both pilot sites (UP and CSS). To ensure that the data analysis in both Italian regions followed the same procedure, we used the same template to report on the analysis and the interviews were fully transcribed (Figure 2).

Concerning sections A–D, all the analyses were conducted with the R Ver. 4.1.2. Statistical software package [The R Project for Statistical computing; available at URL http://www.r-project.org/ (accessed on 14 September 2021)].

For dichotomous variables, differences between the groups were tested using the Fisher exact test. This analysis was made using the 2-Way Contingency Table Analysis available at the Interactive Statistical Calculation Pages [http://statpages.org/ (accessed on 15 September 2021)]. For continuous variables, normal distribution was verified by the Shapiro–Wilk normality test and the 1-sample Kolmogorov–Smirnov test. For normally distributed variables, differences among the groups were tested by the Welch 2-sample t-test or analysis of variance under a general linear model. For non-normally distributed variables, differences among the groups were tested by the Wilcoxon rank-sum test with continuity correction or the Kruskal–Wallis rank-sum test. In the case of a significant Kruskal–Wallis test, an adjustment for multiple comparisons was estimated by Dunn’s tests. Chi-squared analysis was used to compare all frequencies. Test results in which the *p*-value was smaller than the type 1 error rate of 0.05 were declared significant.

Section E was analyzed using the method of Thematic Content Analysis (TCA) [20]. The first level of coding was meant to identify themes and units of meaning. In this, we stayed close to the wording used by the respondent. In the second level of coding, we used more theoretical words. Finally, the third level of coding was the actual analysis: looking for recurring themes, coherence and unique cases. In particular, the TCA is divided into two fundamental analyses: (1) Analysis of vertical content (coding and categorizing by an intra-interview reading (progress of an interview on all codified themes)), and (2) Analysis of horizontal content (second coding and categorizing on reading inter-interviews (illustration of a theme by all the interviews)).

## 3. Results

In this paragraph, all interview section outcomes are described. All tables are included in the Appendix A.

The characteristics of all participants are summarized in Appendix A, respectively, recruited in UP and CSS (respectively, Tuscany and Apulia pilots).

In UP (Appendix A), 36 participants were recruited. Of them, 12 were older persons, 12 were informal caregivers, 8 were professional caregivers, and 4 were social operators. Regarding older persons, 90% of them were with physical frailty, and 50% were with mild cognitive frailty (MMSE = 25.50 ± 1.17 in mean). Informal caregivers cared for their relatives on average 95.75 ± 120.00 months. The average time of day care that informal caregivers provided for their relatives was 6.50 ± 6.72 h. Regarding professional caregivers, the time spent dealing with older adults with physical and/or cognitive frailty was on average 14.40 ± 13.98 years. The participant groups did not differ in sex distribution (*p* = 0.454). Professional caregivers were younger (*p* < 0.0001) with a higher educational level (*p* = 0.003) than other participant groups.

In CSS (Appendix A), 25 participants were recruited. Of them, 10 were older persons, 10 were informal caregivers, and 5 were professional caregivers. Older people were without physical and cognitive (MMSE = 28.08 ± 1.18 in mean) frailty. Informal caregivers cared for their relatives in mean 14.90 ± 17.23 months. The average timeof day care informal caregivers provided for their relatives was in mean 6.50 ± 6.72 h. Regarding professional caregivers, time spent dealing with older adults with physical and/or cognitive frailty was in mean 7.20 ± 6.18 years. The participant groups did not differ in sex distribution (*p* = 0.535) and educational level (*p* = 0.069). Professional caregivers were younger (*p* = 0.002) than other participant groups.

### 3.1. Attitudes towards the Technologies

Comparing the two Italian pilot sites, no significant differences were identified concerning attitudes towards the technologies to be used in the PHArA-ON project from older adults (Appendix A) and all caregivers (Appendix A). Most of the senior citizens declared that they frequently use computers/smartphones/tablets (UP = 83.3%; CSS = 80.0%), and their relatives and friends frequently use computers/smartphones/tablets (UP = 83.3%; CSS = 100.0%). Regarding the utility of the technology, the older adults stated robot, Internet of Things (IoT), app and mobile devices and wearable sensors are more useful (scores ranged from 1 to 3) than other technologies such as artificial intelligence and virtual/augmented reality (scores ranged from 4 to 7). Regarding technology goals, older adults stated that the primary function of a robot could include performing care and assistance tasks (UP = 58.3%; CSS = 40.0%), and its secondary function could include helping family members (UP = 41.7%; CSS = 30.0%); the primary function of IoT could include improving self-care for UP (36.4%); and staying in communication with others for CSS (50.0%), and the secondary function could include helping family members for UP (33.3%), helping operators in their work for CSS (30.0%) and staying in communication with others (UP = 25.0%; CSS = 30.0%); the primary function of AI could be performing care and assistance tasks (UP = 41.5%; CSS = 30.0%), and the secondary function could be improving self-care (as well as to reiterate “performing care and assistance tasks”) for UP (25.0%) and helping operators in their work for CSS (40.0%); the primary function of app and mobile devices could be to provide social opportunities (UP = 27.3%; CSS = 30.0%), and their secondary function could be helping operators in their work and family members for UP (41.7%) and staying in communication with others for CSS (30.0%); the primary function of wearable sensors could be to improve self-care (UP = 36.4%; CSS = 40.0%), and their secondary function could be helping family members (UP = 33.3%; CSS = 30.0%); the primary function of virtual/augmented reality could be performing care and assistance tasks for UP (58.3%) and helping operators in their work for CSS (40.0%), and its secondary function could be helping family members for UP (41.7%) and providing social opportunities for CSS (50.0%).

According to the feedback of all caregivers (except two UP social operators who did not answer the questions in this section), they stated that they frequently use computers/smartphones/tablets (UP = 100.0%; CSS = 100.0%), and their relatives and friends frequently use computers/smartphones/tablets (UP = 100.0; CSS = 100.0%). Regarding the utility of technology, caregivers stated IoT and wearable sensors are more useful (scores ranged from 1 to 3) than other technologies such as robots, artificial intelligence, apps and mobile devices and virtual/augmented reality (scores ranged from 4 to 7). Regarding technology goals, caregivers stated that the primary function of a robot could include performing care and assistance tasks for UP (52.6%) and helping operators in their work for CSS (46.7%), and its secondary function could be helping family members (UP = 33.3%; CSS = 26.7%); the primary function of IoT could be helping operators in their work for UP (40.0%); and improving self-care for CSS (26.7%), and its secondary function could be staying in communication with others (UP = 33.3%; CSS = 46.7%); the primary function of AI could be to help operators in their work for UP (40.0%) and to perform care and assistance tasks for CSS (33.3%), and the secondary function could be to improve self-care (UP = 40.0%; CSS = 26.7%); the primary function of app and mobile devices could be staying in communication with others (UP = 31.6%; CSS = 46.7%), and the secondary function could be improving self-care (as well as to reiterate “staying in communication with others”)for UP (15.0%) and providing social opportunities for CSS (40.0%); the primary function of wearable sensors could be improving self-care (UP = 44.4%; CSS = 46.7%), and the secondary function could be helping family members for UP (38.9%) and performing care and assistance tasks for CSS (33.3%); the primary function of virtual/augmented reality could be helping operators in their work (UP = 38.9%; CSS = 26.7%), and the secondary function could be performing care and assistance tasks for UP (33.3%) and staying in communication with others for CSS (33.3%).

### 3.2. Opinions on the Usefulness of Support Devices

Comparing the two Italian pilot sites, no significant differences were identified concerning opinions on the usefulness of support devices (robot, sensors, etc.) from older adults (Appendix A), except for the CQ5 question (“Devices for performing a Comprehensive Geriatric Assessment” [21]) which resulted in statements such as “very useful” for UP and “mildly useful” for CSS (UP = 72.7%; CSS = 70.0%, *p* = 0.015). Regarding Section C questions, the two pilot sites stated “very useful” CQ1—“Devices for monitoring rest and movements”—(UP = 83.3%; CSS = 90.0%), CQ2—“Devices for monitoring taking medications”—(UP = 83.3%; CSS = 80.0%), CQ3—“Devices for monitoring environmental conditions”—(UP = 66.7%; CSS = 100.0%), CQ4—“Devices for regulating the heating, humidity, lighting, TV channels”—(UP = 66.7%; CSS = 60.0%), CQ6—“Devices for connecting to care programs”—(UP = ns; CSS = 80.0%), CQ7—“Devices for monitoring physical impairment”—(UP = 33.3%; CSS = 90.0%), CQ8—“Devices for monitoring cognitive impairment”—(UP = 41.7%; CSS = 80.0%), CQ9—“Devices for keeping in touch with friends and family”—(UP = 91.7%; CSS = 80.0%), and CQ10—“Bus/taxi service to promote the mobility of the older adults within the city”—(UP = 66.7%; CSS = 80.0%).

Comparing the two Italian pilot sites, no significant differences were identified concerning opinions on the usefulness of support devices (robot, sensors, etc.) from caregivers (Appendix A), except for CQ2 (“Devices for monitoring taking medications”), which resulted in statements such as “very useful” for UP and “mildly useful” for CSS (UP = 81.8%; CSS = 53.3%, *p* = 0.030). Regarding other Section C questions, the two pilot sites stated “very useful” CQ1—“Devices for monitoring rest and movements”—(UP = 77.3%; CSS = 66.7%), CQ3—“Devices for monitoring environmental conditions”—(UP = 77.3%; CSS = 60.0%), CQ5 question—“Devices for performing a Comprehensive Geriatric Assessment”—(UP = 59.1%; CSS = 46.7%),CQ6—“Devices for connecting to care programs”—(UP = ns; CSS = 73.3%), CQ7—“Devices for monitoring physical impairment”—(UP = 72.7%; CSS = 86.7%), CQ8—“Devices for monitoring cognitive impairment”—(UP = 77.3%; CSS = 73.3%), and CQ9—“Devices for keeping in touch with friends and family”—(UP = 72.7%; CSS = 60.0%).

Regarding CQ4 (“Devices for regulating the heating, humidity, lighting, TV channels”) the two pilot sites stated “mildly useful”—(UP = 54.5%; CSS = 60.0%). For CQ10 question (“Bus/taxi service to promote the mobility of the older adults within the city”) the two pilot sites stated, “very useful” and “mildly useful” (respectively, UP = 45.5–50.0%; CSS = 53.3–53.3%).

### 3.3. Impact of PHArA-ON

After showing presentation slides for the identified services, the participants were asked to what extent PHArA-ON can be useful. In the two pilot sites, no significant differences were identified concerning all questions (Appendix A).

Older adults stated “very useful” DQ1—“Improving quality of life”—(UP = 66.7%; CSS = 90.0%), DQ2—“Improving quality of care”—(UP = 91.7%; CSS = 70.0%), DQ3—“Improving safety in daily living activities”—(UP = 91.7%; CSS = 100.0%), DQ4—“Sending emergency alert/communication messages”—(UP = 91.7%; CSS = 90.0%), DQ5—“Improving the assistance provided with physical and cognitive rehabilitation programs at home”—(UP = 58.3%; CSS = 50.0%), DQ6—“Detecting when a person is becoming more lonely and isolated”—(UP = 58.3%; CSS = 90.0%), and DQ7—“Detecting changes in health status”—(UP = 83.3%; CSS = 90.0%).

Caregivers stated “very useful” DQ1—“Improving quality of life”—(UP = 72.7%; CSS = 86.7%), DQ2—“Improving quality of care”—(UP = 77.3%; CSS = 86.7%), DQ3—“Improving safety in daily living activities”—(UP = 81.8%; CSS = 73.3%), DQ4—“Sending emergency alert/communication messages”—(UP = 90.9%; CSS = 93.3%), DQ5—“Improving the assistance provided with physical and cognitive rehabilitation programs at home”—(UP = 72.7%; CSS = 80.0%), DQ6—“Detecting when a person is becoming more lonely and isolated”—(UP = 54.5%; CSS = 86.7%), and DQ7—“Detecting changes in health status”—(UP = 72.7%; CSS = 93.3%).

### 3.4. Users’Needs and Priorities

In Table 3, the prioritization analysis of services according to the stakeholder’s needs is shown.

Through the knowledge/detection of health and personal data, it is possible to activate two fundamental need categories: heath management and socialisation.


➢The specific needs for heath management are the following:
▪Stimulation (high priority): This could improve the assistance provided with home physical and cognitive rehabilitation programs. It could fill a gap due to the lack of operators who can perform this type of care at home and reduce the number of times the patient has to go to the hospital for cognitive stimulation. Programs could be tailored to the clinical condition of each older person. This service will stimulate older adults to engage in a healthier lifestyle by referring to good practices on a daily basis. The devices should be very usable by the patients: “for example, if physical exercises are proposed, the screen must be large, you must see well and hear well, and maybe even… it would be fantastic if, in the face of an exercise that is requested, the device itself also gives feedback on whether it was done well…” (CSS-P01); “It would make possible to maintain physical and cognitive training” (CSS-E04); “It would help me to do physical exercise with the help of a robot or be stimulated with exercises” (UP-E03). Concerning emotional goals, older adults will be involved and empowered. The technical requirements are the following: (1) The system should propose a selection of physical and cognitive exercises that the users can access on their smart TV and/or personal device; (2) The caregiver can select the exercises for the users on the system; (3) The system will analyse the statistics related to the number and performance of exercises completed.▪Monitoring (high priority): Both the doctor and family members could know the health status of the older adult (be aware). This could improve pharmacological compliance and monitor physical exercise and nutrition with a consequent reduction in the workload of caregivers. It will allow older persons to receive remote psychological support in a timelier manner. It will be possible to detect a consequent state of empowerment, safe and reassured. “[…] therefore, the sensors could monitor the conditions and allow the family member to go away for daily tasks. The robot could also have video surveillance functions” (UP-I04); “The safety devices could help the older adult in their autonomy as well as the technologies for the administration of drugs; but also, the equipment for monitoring and implementing individualized care plans” (UP-P03); “It would anticipate hospital practices” (CSS-E07). The technical requirements are the following: (1) Through the Smartphone, Tablet or Smart TV user interface, the older person should be able to videoconference the doctor (or nurse); (2) The system should monitor patients’ total movement in real-time, using data collected by a Smart watch; (3) The aggregate data should be visualized on the caregiver interface; (4) The caregiver should be able to access, over the caregiver interface, the “history” of total movements performed throughout the day.▪Monitoring the environment (medium priority): Monitoring of environmental conditions (through systems for controlling temperature, gas-smoke, light, humidity, entrance-exit of the main doors, etc.), as well as improving safety and well-being, will allow a better organization of care and support. For example, by detecting an ambient temperature that is too high in the home, the system could suggest that the older adult drink more water to hydrate and avoid dehydration (which is one of the most common causes of mortality, particularly frequent in the summer season). All this will lead to a reduction in stress for the older adult and the caregiver, and a reduction in the workload of the latter. It will be possible to allow a consequent state of safety and reassurance. “More than anything else, the control (monitoring of the environment) could create the optimal situation for the safety of the person” (UP-I03). The technical requirements are the following: (1) Some environmental parameters such as humidity, temperature could be measured by the sensors installed in the system; (2) Movement sensors are requested to monitor the total movement within a specific environment; (3) The system should integrate all the sensors and make the collected data available on the caregiver dashboard; (4) Parameters’ pattern analysis (Anomalies detection) was requested.▪Emergency call (low priority): Activation of an emergency call that allows both to intervene in emergencies on time and to contact family members. All this would avoid accidents. The older adults will be remotely supervised 24 h a day and there will be an interaction between the older adults, the referring medical centre and family members, granting everyone a state of reassurance. “Alarm sensors. Screen with video explanation, connection with external via video camera” (UP-P08).➢For Socialisation, the specific need is “To promote social inclusion” with a high priority:
▪To reduce isolation and promote socialisation, a fundamental brick is necessary (knowledge of the person’s history and what his hobbies and habits are).▪Prevent loneliness: “The system should be able to detect an effective phenomenon of isolation, which is often difficult to intercept if we see the patient once every six months” (CSS-P01). Furthermore, it should be more reliable than the observation and the story of the same individual or some family member. With or without the presence of isolation, the system tries to promote the connection with family and friends through participation in social groups or through the reporting of events (selected according to the interests of the individual person) in which to participate, of which the older adult is not aware of.▪Information: The service should send information useful for health. “For example, during seasonal flu, send specific messages on that topic. Or, messages/advice every 4/5 days like << please go out because there are so many activities you can do outdoors rather than at home… move more>>” (CSS-P02). It would be useful to periodically send information to the elderly to keep the communication channel alive with the project and with a structure that follows the project. In addition, the elderly may receive reports of activities in which they can participate, in which they could acquire a new intellectual and/or manual ability through courses that teach them to do something more than what they already know.


## 4. Discussion

In the present manuscript, using a small sample of older adults and informal and professional caregivers, the focus is mainly on the PHArA-ON system to design and evaluate technology in the context of healthy and active ageing. The main goal of this work was to detect and improve the perception, acceptance and usability of technology by end-users.

Even if it is too early to draw outcomes and conclusions about the application of these technologies in clinical practice, we obtained useful insights about the attitude and availability of end-users in using them.

Indeed, these dataare of great importance, since they not only present useful indications to assess what has been accomplished up to now, but alsoprovide important guidelines to improve the system, while specific experimentation stages are expected to be carried out over the next months. Indeed, researchers could take advantage of these studies to understand which kind of technology is more acceptable because it was rated higher for a certain application. Moreover, they could also obtain feedback regarding the development priority related to the impact of the proposed services.

Current studies have highlighted the importance of acquiring relevant knowledge on user needs to develop technologies that can handle the real life situations of the older people [22,23]. When developing technological products, it is essential to consider social dynamics and contextual factors [24,25] in order to increase the usability and interaction with technological devices.

In this respect, the design of technological products should address a deep inclusion of older people during the development process. The PHArA-ON project deals with this aspect. Indeed, the mission of the project is to develop advanced technological solutions for ageing by defining, developing and demonstrating an interactive development process with end-users, so as to design specific guidelines for the implementation of the outlined technological services.

In our study (Table 4), there are some different characteristics between the two pilot sites participants; they seem to have almost the same opinion about the technology that could support their needs and what they expect from the PHArA-ON project. Concerning the attitude towards the technologies, they seem to have good knowledge and skill in the use of computer/smartphone/tablet. Older adults and caregivers agree on the utility of IoT and wearable sensors, in line with other studies [26,27,28]. Moreover, most of the participants (in line with previous studies) agree with the primary function of robots and AI to perform care and assistance tasks [29,30,31], IoT and wearable sensors to improve self-care [26], apps and mobile devices to have social opportunities [32], and virtual/augmented reality to help operators in their work [33]. Regarding the secondary function, a higher percentage of participants (in line with previous studies) agree with the utility of robots and wearable sensors to help family members [26,29,30], IoT, app and mobile devices to stay in communication with others [26,32], AI to improve self-care [31], virtual/augmented reality to have social opportunities and staying in communication with others [33]. Older adults and caregivers think that support devices (robot, sensors, etc.) are very useful to monitor rest and movements, taking medications, environmental conditions, physical and cognitive impairment, perform a Comprehensive Geriatric Assessment (CGA), connect to care programs and keep in touch with friends and family. In a similar study, it was reported that technological systems could be very useful to improve quality of life, care and safety, monitoring bed rest and movements, medication use, and ambient environmental conditions, and emergency communication [34].

After viewinga presentation about previously identified scenarios and services, elderly and caregivers think that the PHArA-ON system could be very useful to improve quality of life, quality of care, safety in daily living activities, the assistance provided with physical and cognitive rehabilitation programs at home, sending emergency alert/communication messages, detecting when a person is becoming lonelier or isolated and changes in health status. The aforesaid outcomes can be sustained by a previous study, which showed that technology may be a useful tool in mitigating depression and loneliness, while enhancing social connectedness, resilience, and overall quality of life for people with dementia [35].

Through the prioritization analysis of services according to the stakeholder’s needs, two fundamental need categories have been determined: Health Management and Socialisation. For Heath Management, the specific needs with a high priority are Stimulation and Monitoring, with a medium priority of Monitoring the environment, and a low priority of Emergency call. For Socialisation, the specific need is “To promote social inclusion” with a high priority, which includes reducing isolation and promoting socialization, stopping loneliness and information. Previously, other studies have reported that technological service platforms could develop professional social work vigorously and cooperate with hospital centres in order to address the problem of social isolation of older adults, improve their physical and mental health, as well as quality of life, and promote the healthy aging of the population [36,37].

The capability of technology to improve the lives of older subjects appears a feasible objective to reach as soon as possible, considering the demographic shift, the economic constraint, and societal changing [38]. The use of these technologies could be considered from the point of view of care professionals as an interesting opportunity to save more time to devote to patient care. This project also has the added value of defining a standardized methodological approach for the development of new technologies that could improve their faster implementation and industrialization.

Moreover, including cognitive support in the design of technological products for domestic assistive services can be more useful in terms of independent living both at home and in the community. Further research could be run in order to confirm the usefulness of interviews and/or questionnaires concerning the real needs of end-users in the first phase of technological system development.

Limitations of the present study should also be considered when interpreting our findings. In particular, in UP, the elderly had physical and cognitive impairment, while in CSS, elderly did not have physical and cognitive impairment; informal caregivers in UP cared for their relatives for much longer than CSS; professional caregivers in UP, dealing with the elderly with physical and/or cognitive frailty for much longer than CSS. UP is a social enterprise including a network of ten social cooperatives where patients are followed for a long time for in-home support, while CSS is a hospital where older adults were recruited in a geriatric evaluation unit for the interview.

## 5. Conclusions

Our analysis represented a point of crucial importance, not only in developing and improving the system by taking into consideration end-users’ (both patients and caregivers) expectations and needs, but also in leading to the development of a prototype and the experimentation stage as well.

The following are the main scientific contributions of this study:-To design and evaluate technology in the context of healthy and active ageing;-To acquire relevant knowledge on user needs to develop technologies that can handle real life situations of older people;-Useful insights about the attitude and availability of end-users in using technologies in clinical practice;-To provide important guidelines to improve the PHArA-ON system while specific experimentation stages are expected to be carried out;-To understand which kind of technology is more acceptable;-To provide feedback regarding the development priority related to the impact of the proposed services.

In conclusion, this work, through fruitful and continuous interaction with different subjects involved in the process of the development of the system, as well as stakeholders, will enable the implementation of a platform, which could be further and easily integrated and improved.

## Figures and Tables

**Figure 1 sensors-22-00163-f001:**
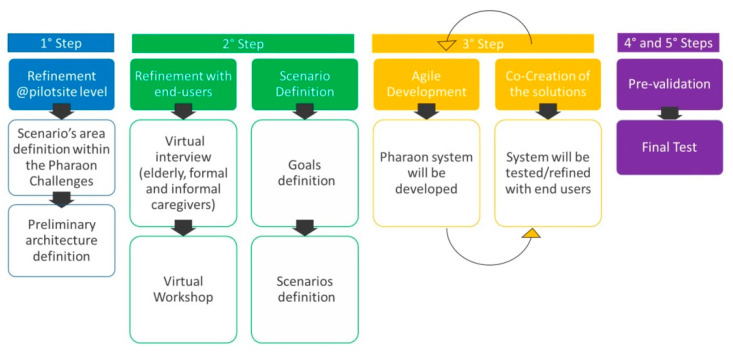
Overview of the PHArA-ON methodology applied at Italian Pilot level.

**Figure 2 sensors-22-00163-f002:**
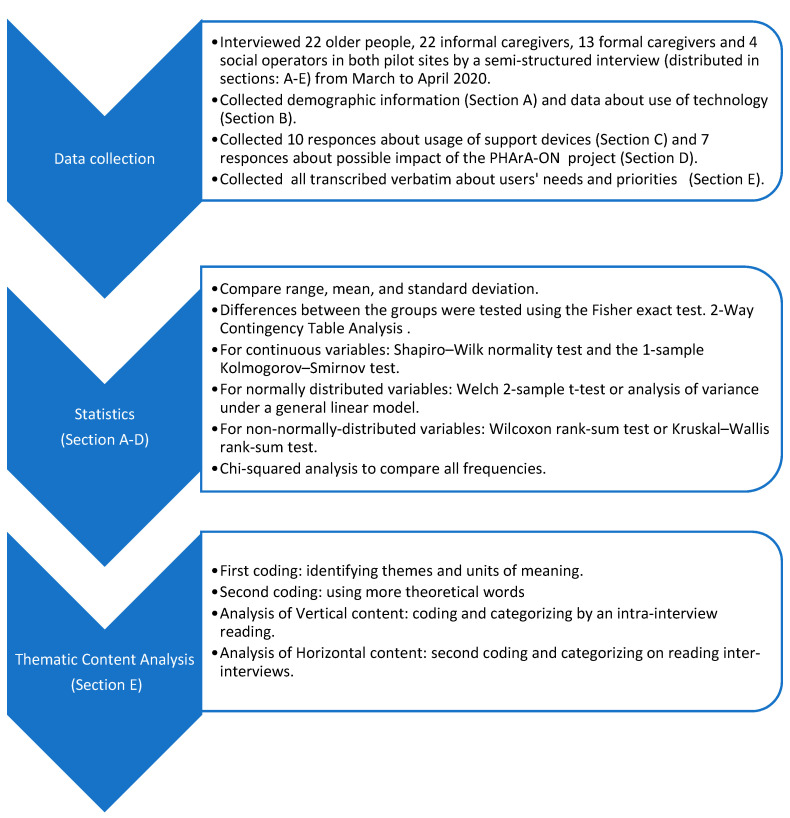
Flow chart regarding the analysis phases carried out.

**Table 1 sensors-22-00163-t001:** Technologies customized to fulfil services in specific intervention areas.

Area of Intervention	Description of Service	System Components
Remote monitoring	✓ Sensors could monitor movements; ✓ Dedicated algorithms should analyse the data and identify potential suspicious situation; ✓ The system could activate an “emergency call” that allows carers to promptly intervene.	Automatize the Comprehensive Geriatric Assessment (CGA) monitoring service; Wearable devices and sensors—to monitor the health status of patients and the parameters of the CGA; Interfaces which manage data acquisition and data visualization; Telepresence robot to move remotely in the environment to check the status.
Facilitating socialization	✓ PHArA-ON interfaces could keep in touch with the other actor of the care process. ✓ PHArA-ON app could allow to chat, organize visits, virtually meet and play cards.	Smart TV and PHArA-ON application that manages socialisation people of the same community. Telepresence robot at home which could promote the interaction, proposing telco with relatives but also with clinicians and psychologists to reduce the social isolation. Platform where older people can be informed about local news and events.
Cognitive and physical stimulation program	✓ Personalized physical/cognitive exercises will be performed at home. ✓ Caregiver will analyse the data collected and personalized the plan.	Personalized interface where people can access and visualize important data (e.g., Smart TV or tablet). Smart solutions to promote personalized care plan including physical and cognitive stimulation (i.e., through games). Smart algorithms for personalization of the care plan.
Support caregiver at work	✓ Professional caregivers could access the interface to monitor the patient’s health status, providing feedback to personalize the care. ✓ Informal caregivers will receive in their own devices (smartphone, tablet, etc.) the information about health and social status of the elders.	Integrated Care platforms which will manage the connection between the devices and the management of data. Smart AI algorithms that analyse aggregated data.

**Table 2 sensors-22-00163-t002:** Overview of system components.

Service	System Components
Managing and storing health data	Integrated Care platforms which will manage the connection between the devices and the management of data. Smart AI algorithms that analyse aggregated data.
Remote monitoring	Robot which could automatize the Comprehensive Geriatric Assessment (CGA) monitoring service; Wearable devices and sensors—To monitor the health status of the patients and the parameters of the CGA. Interfaces which manages the data acquisition, the test administration and the data visualization.
Facilitating interactions	Smart TV and PHArA-ON application that manages the social meeting among peoples of the same community. Telepresence robot at home which could promote the interaction proposing telco with relatives but also with clinicians and psychologist to reduce social isolation.
Personalized service at seniors’ home and in the domestic environment	Personalized interface where people can access and visualize important data (i.e., include a Smart TV). Smart solutions to promote personalized care plan including physical and cognitive stimulation (i.e., through games) and teleassistance; Smart algorithms for decision support.

**Table 3 sensors-22-00163-t003:** Prioritization Analysis of services according to needs of the stakeholders.

Need Category	Specific Need	Activities	Emotional Goals	O	I	P	Final Priority	Technical Requirement	Technologies
**Manage Health**	Emergency call	1. Make an emergency call.	Reassured Safe	0	2	3	Low (5)	The system sends Email, messages or pop-up to family and/or caregivers if anomalies are detected.	Caregivers’ Tablet or Smartphone; Caregiver’s interface; Anomalies detection system
								Older adults will use the app to make an alert call to the caregiver.	Older person’s Tablet or Smartphone; Robot; Caregiver’s interface; Older adult’s interface
	Stimulation	1. The older adult is stimulated to perform physical exercises.	Involved	2	5	6	High (13)	The older person should be able through the Smart TV (browser or app) to selectphysical exercises.	Smart TV; Older adults’ Tablet or Smartphone Caregiver’s Tablet or Smartphone Older adult’s interfaceCaregiver’s interface; Physical exercises system
								The caregiver can select the exercises for the users on the system.	
								The system will analyse the statistical related to the number of exercises performed.	
								The system will analyse the performance of the exercise.	
		2. The older adult is stimulated to perform cognitive exercises.	Empowered	3	5	6	High (14)	The system should propose a selection of cognitive exercises that the users can access on his Smart TV and/or personal device.	Smart TV; Older adult’s Tablet or Smartphone Caregiver’s Tablet or Smartphone Cognitive exercises systemCaregiver’s interface
								The caregiver can select the exercises for the users on the system	
								The system will analyse the statistical related to the number of exercises performed.	
								The system will analyse the output of the cognitive stimulation/assessment performed.	
	Monitoring	1. Psychological support.	Reassured Empowered	6	4	3	High (13)	Through the Smart TV user Interface the older person should be able to call in videoconference the doctor (or nurse).	Smart TV Older adult’s Tablet or Smartphone; Older adult’s interface; Caregiver’s Tablet or Smartphone
								Through the Smartphone or Tablet the older person should be able to call in videoconference the doctor (or nurse).	
		2. The caregiver monitors physical activities of older adult.	Safe	3	5	3	High (11)	The system should monitor patients’ total movement in real time, by means of data collected by Smartwatch.	SmartwatchCaregiver’s Tablet or Smartphone Caregiver’s interface; Analysis system
								The aggregate data should be visualized on the Caregiver interface.	
								The caregiver should be able to access over the Caregiver interface also to the “history” of total movements performed over the day.	
		3. Monitoring the environment	Reassured Safe	3	5	2	Medium (10)	Some environmental parameters like Humidity, Temperature are measured by the sensors installed in the system.	Sensors; Caregiver’s Tablet or Smartphone Caregiver’s interfaceAnalysis system
								PIR movement are requested to monitor the total movement in a space.	
								The system should integrate all the sensors and make the collected data available on the caregiver dashboard.	
								Parameters’ pattern analysis (Anomalies detection).	
**Socialize**	To promote social inclusion	1. Support by the informal caregiver in inserting information about patient’s life	Empowered	6	5	3	High (14)	The Smart TV will show information about his hobbies. Additionally, he can also see some old photos.	Smart TV Smartphone or tablet
								The older person can insert his photo in the system.	
		2. Maintaining relationships with friends	Involved					The older should call and see his friends by using the television.	Smart TV Older adult’s Tablet or Smartphone Older adult’s interface
								The older can create on his interface small groups of friends to video call.	
		3. Information	Involved Cared of					The system should give advice on local events such as contain courses and news about good health practices.	Smart TV

Legend: O, Older people; I, Informal caregiver; P, Professional caregiver and social operators.

**Table 4 sensors-22-00163-t004:** Summary table of the results obtained and comparison with respect to previous studies.

	Older Adults		Caregivers		Aligned with Previous Studies [Ref.]
	UP	CSS	UP	CSS	
Technology goal:					
- Robot					[21,22,23,24,25,26,27,28]
To perform care and assistance tasks	58.30%	40.00%	52.60%	-	
Helping operators in their work	-	-	-	46.70%	
Helping family members	41.70%	30.00%	33.30%	26.70%	
- IoT					
Improving self-care	36.40%	-	-	26.70%	
Staying in communication with others	-	50.00%	-	-	
Helping family members	33.30%	-	-	-	
Helping operators in their work	-	30.00%	40.00%	-	
Staying in communication with others	25.00%	30.00%	33.30%	46.70%	
- AI					
Performing care and assistance tasks	41.50%	-	-	33.30%	
Improving self-care	25.00%	40.00%	40.00%	26.70%	
Helping operators in their work	-	30.00%	40.00%	-	
- App and mobile devices					
Having social opportunities	27.30%	30.00%	-	40.00%	
Helping operators in their work and family members	41.70%	-	-	-	
Staying in communication with others	-	30.00%	31.60%	46.70%	
Improving self-care	-	-	15.00%	-	
- Wearable sensors					
Improving self-care	36.40%	40.00%	44.40%	46.70%	
Helping family members	33.30%	30.00%	38.90%	-	
Performing care and assistance tasks	-	-	-	33.30%	
- Virtual/augmented reality					
Performing care and assistance tasks	58.30%	-	33.30%	-	
Helping operators in their work	-	40.00%	38.90%	26.70%	
Helping family members	41.70%	-	-	-	
Having social opportunities	-	50.00%	-	-	
Staying in communication with others	-	-	-	33.3%).	
Support devices to:					[29]
- Monitor rest and movements	83.30%	90.00%	77.30%	66.70%	
- Taking medications	83.30%	80.00%	81.80%	51.80%	
- Environmental conditions	66.70%	100.00%	77.30%	60.00%	
- Physical/cognitive impairment	33.3%/41.7%	90.0%/80.0%	72.7%/77.3%	86.7%/73.3%	
- Perform a Comprehensive Geriatric Assessment	-	-	59.10%	46.70%	
- Connect to care programs	-	80.00%	-	73.30%	
- Keep in touch with friends and family	91.70%	80.00%	72.70%	60.00%	
Impact of PHARA.On on:					[30]
- Improving quality of life	66.70%	90.00%	72.70%	86.70%	
- Improving quality of care	91.70%	70.00%	77.30%	86.70%	
- Improving safety in daily living activities	91.70%	100.00%	81.80%	73.30%	
- Sending emergency alert/communication messages	91.70%	90.00%	90.90%	93.30%	
- Improving the assistance	58.30%	50.00%	72.70%	80.00%	
- Detecting when a person is lonely and isolated	58.30%	90.00%	54.50%	86.70%	
- Detecting changes in health status	83.30%	90.00%	72.70%	93.30%	
	**Need category**	**Specific needs**	**Priority level**	**Emotional goal**	
Users’ needs and priorities	Manage health	Stimulation	High	Involved	[33]
		Monitoring	High	Reassured	[33]
	Socialize	To promote social inclusion	High	Empowered	[31,32]

## Data Availability

The data presented in this study are available on request from the corresponding author. The data are not publicly available due to restrictions (their containing information that could compromise the privacy of research participants).

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
