# Peer review of "Pilots for Healthy and Active Ageing (PHArA-ON) Project: Definition of New Technological Solutions for Older People in Italian Pilot Sites Based on Elicited User Needs"

_sensors, 2021, doi:10.3390/s22010163_

Round 1
Reviewer 1 Report
I think the article has some aspects to improve:
Major concerns:
1.- The authors indicated that "The aim of the present study was to provide a quantitative and qualitative analysis about 76 real needs of older people and their caregivers and to identify the priority needs that 77 should lead development of PHArA-ON system and its customizable interoperable open 78 platforms with advanced services, devices and technological tools". However, they did not indicate explicitly what is the problem that they want to solve or face. Please establish the main problem that this research addresses.
2.- Is the questionnaire valid? This is something very important to keep in mind for the study. The questionnaire have to be valid (e.g. construct validity, content validity, face validity), reliable (internal consistency), responsive, among others. Please provide evidence in your article on these aspects and others related to a good questionnaire.
3.- a) What is the exact name of the ethics committee?
b) What is the exact approval number that the ethics committee gave you?
4.- It is difficult to know what is the contribution of the article. I think you should explicitly state this in the article.
Minor concerns:
5.- The tables presented in section 3 are very long. Can you move them to an appendix or supplementary material file?
6.- The following reference has no number: "Blinka MD, Buta B, Bader K, Hanley CL, Schoenborn N, McNabney M, et al. Developing a sensor-based mobile application for 952 in-home frailty assessment: a qualitative study. Innov Aging. 2019;3(Supplement_1):S831–S8S2."
7.- I think the appendix A should be moved to a supplementary material file.
8.- In the "Author contributions" you will find the author F.Ci. instead of F.C.
Author Response
Reviewer 1
I think the article has some aspects to improve:
Major concerns:
1.- The authors indicated that "The aim of the present study was to provide a quantitative and qualitative analysis about real needs of older people and their caregivers and to identify the priority needs that should lead development of PHArA-ON system and its customizable interoperable open platforms with advanced services, devices and technological tools". However, they did not indicate explicitly what is the problem that they want to solve or face. Please establish the main problem that this research addresses.
- In Introduction section, it is already described that “PHArA-ON is focusing on setting up and merging Health and Care at home for older vulnerable subjects or moderately frail individuals. In doing so, the pilot wants to promote correct lifestyles (e.g., characterized by personalized diets and physical exercise programs, social connectivity) and health status monitoring at home”. These sentences clear the problem to solve (frailty) and the actions to propose in order to manage the problem (correct lifestyles and health status monitoring at home).
2.- Is the questionnaire valid? This is something very important to keep in mind for the study. The questionnaire have to be valid (e.g. construct validity, content validity, face validity), reliable (internal consistency), responsive, among others. Please provide evidence in your article on these aspects and others related to a good questionnaire.
- In the Material and Methods section, it is already described that “The interview was first pretested with 20 older people who were representative of the Italian population in terms of gender, language, and age. Some small adjustments were made to the questionnaire following this step”.
3.- a) What is the exact name of the ethics committee?
b) What is the exact approval number that the ethics committee gave you?
3. We added the exact name of ethics committee and the approval number.
4.- It is difficult to know what is the contribution of the article. I think you should explicitly state this in the article.
- In Discussion section, we described the following concepts: “These data, indeed, are of great importance since they not only give useful indications to assess what has been accomplished up to now but also, provide important guidelines to improve the system while specific experimentation stages are expected to be carried out over the next months. Indeed, researchers could take advantage of these studies to understand which kind of technology is more acceptable because it was rated higher for a certain application. Moreover, they could also have feedback regarding the development priority related to the impact of the proposed services”. We reported current studies which highlighted the importance to acquire relevant knowledge on user needs to develop technologies that can handle real life situations of the older people. When developing technological products, it is essential to consider social dynamics and contextual factors in order to increase the usability and interaction with technological devices.
Minor concerns:
5.- The tables presented in section 3 are very long. Can you move them to an appendix or supplementary material file?
- According to reviewer observation, we moved the tables in a supplementary material file.
6.- The following reference has no number: "Blinka MD, Buta B, Bader K, Hanley CL, Schoenborn N, McNabney M, et al. Developing a sensor-based mobile application for 952 in-home frailty assessment: a qualitative study. Innov Aging. 2019;3(Supplement_1):S831–S8S2."
- According to the reviewer observation, we attributed the following reference at the number 10.
7.- I think the appendix A should be moved to a supplementary material file.
- According to reviewer comment, we moved the Appendix A in the supplementary material file.
8.- In the "Author contributions" you will find the author F.Ci. instead of F.C.
- We have entered F.Ci. and F.C. to differentiate the two co-authors Filomena Ciccone (F. Ci.) and Filippo Cavallo (F.C.).
Reviewer 2 Report
Dear Authors:
Using quantitative and qualitative data, the opinion and needs of 22 older adults, 22 informal caregivers, 13 professional caregivers and 4 social operators, regarding the services offered by PHArA-ON project, was explored. In general, it is not a well-organized and well-written manuscript, not reaching the high standards of the Sensors journal. A relevant literature review is missing, the study design is unclear and needs clarifications. The presentation of the results sometimes is not clear. A Conclusion section is missing. The study lacks of scientific rigor and innovation. Main contributions to the field are unclear. Please find below some major/minor comments regarding the submitted manuscript:
Introduction
- A relevant “background” section is clearly missing.
- Change from “Pharaon” to “PHArA-ON”
- A relevant theoretical approach is needed to sustain the proposed study.
- The use of terms “physical exercise” are not properly corrected. The authors should opt by the use of physical activity or/and exercise.
- The following is not clear: “emphasize correct lifestyles”.
Materials
- Reference is missing in the following “guidelines [ ].”
- Please provide the ref. number of the obtained Ethical approval. Was the ethical approval obtained from the 5 different countries? If yes, this info should be clearly presented.
- Change from “composed of” to “composed by”
- The following info should not be in the Materials section: “The results of the two focus groups were summarized in Table 2.”
- The Overall perspective of the H2020 project ““PHArA-ON” should be presented at the beginning to better understand the proposed pilot studies, scenarios and related aims/purposes.
- The use of “co-creative sessions” should be better explained and connected with strong theoretical approach.
- “The attitude toward the selected technology to define whether or not the proposed technology will be used if integrated into the services.”. The authors should be more specific…which technology?
- How long took the interview? When was performed? This info should be presented
- “Some small adjustments were made to the questionnaire following this step.” The authors should be more concrete…Which adjustments were made?
- The study design is not clear
Results
- The verb used in the following sentence should be in the present instead of past: “In this paragraph, all interview section outcomes were described.” To “ …is described.”
- The quantitative results are mere descriptive statistics, based on frequencies and percentages.
- The qualitative results (section 3.4) are presented in a very confusing way. This section should be clearly reviewed.
- The title of section 3.5 is “Tables”?...
Discussion
- The use of questions in this section is inappropriate.
- Scientific writing style is clearly missing.
An independent “Conclusions” section should be added.
Author Response
Reviewer 2
Dear Authors:
Using quantitative and qualitative data, the opinion and needs of 22 older adults, 22 informal caregivers, 13 professional caregivers and 4 social operators, regarding the services offered by PHArA-ON project, was explored. In general, it is not a well-organized and well-written manuscript, not reaching the high standards of the Sensors journal. A relevant literature review is missing, the study design is unclear and needs clarifications. The presentation of the results sometimes is not clear. A Conclusion section is missing. The study lacks of scientific rigor and innovation. Main contributions to the field are unclear. Please find below some major/minor comments regarding the submitted manuscript:
- Introduction
A relevant “background” section is clearly missing.
Change from “Pharaon” to “PHArA-ON”
A relevant theoretical approach is needed to sustain the proposed study. The use of terms “physical exercise” are not properly corrected. The authors should opt by the use of physical activity or/and exercise.
The following is not clear: “emphasize correct lifestyles”.
- According to the reviewer observations, we improved the Introduction section and made the following corrections:
- Changed from “Pharaon” to “PHArA-ON”.
- The theoretical approach to sustain the study clarified in Introduction and Discussion sections.
- Opted by the use of “physical exercise”.
- Changed from “emphasize correct lifestyles” in “promote correct lifestyles”.
- Materials
Reference is missing in the following “guidelines [ ].”
Please provide the ref. number of the obtained Ethical approval. Was the ethical approval obtained from the 5 different countries? If yes, this info should be clearly presented.
Change from “composed of” to “composed by”
The following info should not be in the Materials section: “The results of the two focus groups were summarized in Table 2.”
The Overall perspective of the H2020 project ““PHArA-ON” should be presented at the beginning to better understand the proposed pilot studies, scenarios and related aims/purposes.
The use of “co-creative sessions” should be better explained and connected with strong theoretical approach.
“The attitude toward the selected technology to define whether or not the proposed technology will be used if integrated into the services.”. The authors should be more specific…which technology?
How long took the interview? When was performed? This info should be presented
“Some small adjustments were made to the questionnaire following this step.” The authors should be more concrete…Which adjustments were made?
The study design is not clear
- According to the reviewer observations, we made the following corrections:
- The reference n. 11 was attributed to the observed missing reference.
- The Ethical approval information about the 2 Italian Pilot sites have been reported in the text.
- Changed from “composed of” to “composed by”
- “The results of the two focus groups were summarized in Table 2” and overall perspective of the H2020 project PHArA-ON have moved on Introduction section.
- Clarified the technology proposed during the interview.
- All information about the interview was added.
- The interview adjustments have been clarified in the text.
- Results
The verb used in the following sentence should be in the present instead of past: “In this paragraph, all interview section outcomes were described.” To “ …is described.”
The quantitative results are mere descriptive statistics, based on frequencies and percentages.
The qualitative results (section 3.4) are presented in a very confusing way. This section should be clearly reviewed.
The title of section 3.5 is “Tables”?...
- According to the reviewer observations, we made the following corrections:
- Changed from “were described” to “ are described”.
- Since this is an interview, the results are obviously reported in percentages and frequencies.
- The section 3.4 have been re-organized.
- We moved the tables in a supplementary material file.
- Discussion
The use of questions in this section is inappropriate.
Scientific writing style is clearly missing.
An independent “Conclusions” section should be added.
- According to the reviewer observations, we made the following corrections:
- The questions have been deleted.
- The Discussion section have been improved.
- A Conclusion section have been added.
Reviewer 3 Report
This is a manuscript on determining the needs and preferences of older people 19 and their caregivers for improving the healthy and active aging, and guiding technological development of PHArA-ON system.
The article is well organised and the results are presented clearly. The authors should consider the following comments.
line 173: "data analysis in both countries": Are the results shown in this manuscript coming only from the Italian pilot site (Tuscany and Apulia)?
Line 175: "R Ver. 2.8.1" This version is quite outdated.
Line 186: In the case of a significant Kruskal-Wallis test, did you use an adjustment for multiple comparisons?
Table 5: AI (primary function): To help operators in their work – n (%) -- please check the frequency 43 in UP.
Table 6: Are social caregivers included in the UP sample?
Author Response
Reviewer 3
- This is a manuscript on determining the needs and preferences of older people and their caregivers for improving the healthy and active aging, and guiding technological development of PHArA-ON system.
The article is well organised and the results are presented clearly. The authors should consider the following comments.
- We thank the reviewer for his/her positive comments. According to his/her suggestions, we made the following corrections.
- Line 173: "data analysis in both countries": Are the results shown in this manuscript coming only from the Italian pilot site (Tuscany and Apulia)? 2. Yes, they are. We changed from “countries” in “Italian regions”.
- Line 175: "R Ver. 2.8.1" This version is quite outdated. 3. Correct “R Ver. 2.8.1” in “R Ver. 4.1.2”.
- Line 186: In the case of a significant Kruskal-Wallis test, did you use an adjustment for multiple comparisons? 4. Yes, we did. The sentence “In the case of a significant Kruskal-Wallis test, an adjustment for multiple comparisons have been estimated by Mann-Whitney tests” have been added in the text.
- Table 5: AI (primary function): To help operators in their work – n (%) -- please check the frequency 43 in UP. 5. Correct the typo about aforesaid frequency.
- Table 6: Are social caregivers included in the UP sample? 6. We added in the text: “except two UP social operators who did not answer the questions in this section”.
Round 2
Reviewer 1 Report
Thank you very much to the authors for considering my comments.
In my opinion, the changes made are sufficient for me, and my decision is to accept the article in its present form.
Author Response
We thank the reviewer for his/her support and positive feedback.
Reviewer 2 Report
Dear Authors:
I believe that the manuscript has not been sufficiently improved to reach the high standards of the Sensors journal.
The study lacks of scientific rigor and innovation. Main contributions to the field are unclear.
The use of terms “physical exercise” are not properly used (in R2 they used the terms "physical exercise exercises"; even worse...).
A relevant and strong theoretical framework is needed to sustain the proposed study (e.g., an independent Background section).
Author Response
We thank the reviewer for his/her comments and suggestions in order to improve the manuscript according to the high standards of the Sensors journal.
In Discussion section, we described the following concepts: “These data, indeed, are of great importance since they not only give useful indications to assess what has been accomplished up to now but also, provide important guidelines to improve the system while specific experimentation stages are expected to be carried out over the next months. Indeed, researchers could take advantage of these studies to understand which kind of technology is more acceptable because it was rated higher for a certain application. Moreover, they could also have feedback regarding the development priority related to the impact of the proposed services”.
We reported current studies which highlighted the importance to acquire relevant knowledge on user needs to develop technologies that can handle real life situations of the older people. When developing technological products, it is essential to consider social dynamics and contextual factors in order to increase the usability and interaction with technological devices.
According to reviewer observation, we correct the typo"physical exercise exercises".
In Introduction section, it is already described that “PHArA-ON is focusing on setting up and merging Health and Care at home for older vulnerable subjects or moderately frail individuals. In doing so, the pilot wants to promote correct lifestyles (e.g., characterized by personalized diets and physical exercise programs, social connectivity) and health status monitoring at home”. These sentences clear the problem to solve (frailty) and the actions to propose in order to manage the problem (correct lifestyles and health status monitoring at home).
Reviewer 3 Report
The authors have replied adequately to most of the comments.
There is one remaining issue. Please reconsider the reply on multiple comparison adjustment after a Kruskal Wallis test is found significant. Mann-Whitney U test is not used to adjust/correct for multiple testing but only to examine pairwise comparisons.
Author Response
We thank the reviewer for his/her positive comments. According to his/her suggestions, we made the following corrections: Dunn's test has been used for multiple comparisons.
This manuscript is a resubmission of an earlier submission. The following is a list of the peer review reports and author responses from that submission.
Round 1
Reviewer 1 Report
I have some concerns about this article:
1.- Even though the article has a lot of information and the authors did a study, it is difficult to see this article as a research paper, because I can not find in it a suitable research question or solution to a problem. I think the authors should sort and clarify the ideas presented in this article.
2.- It is not clear how the results of this article can be used by other researchers. This must be clarify in this article.
3.- Please indicate explicitly in the paper the name of the ethics committee that authorized the study and the approval number issued by the committee.
4.- Please explain what is EU H2020-SC1-FA-DTS-2018-2020 call.
5.- The Interview on PHArA-ON services (Appendix A, sections A to F):
a) Was the questionnaire validated?
b) What kind of validation was done?
c) What values were reported?
d) Please incorporate all the answers to the previous questions in the article.
6.- Please explain in the article why the authors use those Recruitment criteria for respondents.
7.- In relation to the tables, I think that the authors can move some of them to a supplementary material file according to the rules of the journal.
8.- Please rewrite the discussion section. A discussion section should compare the results of the study with other studies in the literature. However, the authors just comment about their results without comparison. One or a couple of references is convenient for each paragraph/result.
9.- In relation to "Author Contributions":
a) to the best of my knowledge, some authors do not qualify for authorship according to International Council of Medical Journal Editors (ICMJE), and the rules of the journal. Maybe some information was omitted. Please review this issue.
b) Is author D.G. actually G.D.?